# Advances in Antiretroviral Therapy for Patients with Human Immunodeficiency Virus-Associated Tuberculosis

**DOI:** 10.3390/v16040494

**Published:** 2024-03-23

**Authors:** Xiaoqin Le, Yinzhong Shen

**Affiliations:** Department of Infection and Immunity, Shanghai Public Health Clinical Center, Fudan University, Shanghai 201508, China; 17211300007@fudan.edu.cn

**Keywords:** antiretroviral therapy, HIV infection, tuberculosis, immune reconstitution inflammatory syndrome

## Abstract

Tuberculosis is one of the most common opportunistic infections and a prominent cause of death in patients with human immunodeficiency virus (HIV) infection, in spite of near-universal access to antiretroviral therapy (ART) and tuberculosis preventive therapy. For patients with active tuberculosis but not yet receiving ART, starting ART after anti-tuberculosis treatment can complicate clinical management due to drug toxicities, drug–drug interactions and immune reconstitution inflammatory syndrome (IRIS) events. The timing of ART initiation has a crucial impact on treatment outcomes, especially for patients with tuberculous meningitis. The principles of ART in patients with HIV-associated tuberculosis are specific and relatively complex in comparison to patients with other opportunistic infections or cancers. In this review, we summarize the current progress in the timing of ART initiation, ART regimens, drug–drug interactions between anti-tuberculosis and antiretroviral agents, and IRIS.

## 1. Introduction

The WHO global tuberculosis report of 2023 pointed out that an estimated 10.6 million people fell ill with tuberculosis worldwide in 2022, of which 6.3% were people living with human immunodeficiency virus (HIV) [1] (≈668,000 people). In the individual host, *Mycobacterium tuberculosis* (*M. tuberculosis*) and HIV potentiate one another [2]. HIV co-infection is one of the greatest risk factors for developing active tuberculosis, and *M. tuberculosis* co-infection promotes the progression of HIV infection into AIDS [3]. The intricate interactions have brought big challenges for the diagnosis and treatment of the two diseases. Due to remarkable immune deficiency, people living with HIV (PLWH) with active tuberculosis might present with non-specific clinical symptoms or only with radiology or sputum test changes. Meanwhile, due to fewer bacilli in sputum, positive rates of acid-fast bacilli smear microscopy and culture examination are lower than in HIV-uninfected patients [4]. Additionally, there are several important management challenges for co-treatment of HIV and tuberculosis: (i) when to start antiretroviral therapy (ART); (ii) important drug–drug interactions; (iii) additive toxicities of concomitant antiretroviral (ARV) agents and anti-tuberculosis drugs; and (iv) tuberculosis immune reconstitution inflammatory syndrome (TB-IRIS) [5]. However, ART and tuberculosis treatment are both pivotal for preventing unnecessary deaths and improving survival, as well as to reduce onward transmission of both infections [5]. The 2023 tuberculosis report revealed that among PLWH who were diagnosed with tuberculosis, the combination of tuberculosis treatment and ART is estimated to have averted 6.4 million deaths between 2010 and 2022 [1]. In recent years, lots of remarkable progress and breakthroughs have been made in the ART-related research among HIV-*M. tuberculosis* coinfected people. To assist physicians to make better clinical decisions when facing the concurrent treatment of HIV and tuberculosis, we reviewed the current literature regarding new progress in ART for patients with HIV-associated tuberculosis.

## 2. Epidemiology of HIV-Related Tuberculosis

Globally, in 2022, 7.5 million people were newly diagnosed with tuberculosis and an estimated 1.30 million died of tuberculosis [1]. Five main risk factors are attributed to the new cases of tuberculosis, including undernourishment, HIV infection, alcohol use disorders, smoking (especially among men) and diabetes [1]. The global incidence rate of active tuberculosis in PLWH has steadily declined for the past several years; however, tuberculosis remains the leading cause of mortality among PLWH [1,6,7,8,9] regardless of near-universal access to ART and tuberculosis preventive therapy. In 2022, tuberculosis caused nearly 167,000 deaths in PLWH across the world [1], more than one-fourth of all HIV-related mortalities (630,000 people died from HIV-related illnesses in 2022) [10]. Of the tuberculosis deaths in PLWH, 50% were men, 40% were women and 9.8% were children [9]. HIV-related tuberculosis remains one of the most substantial health issues among PLWH.

*M. tuberculosis* is transmitted through droplet nuclei from the respiratory tract mainly when an infected patient coughs [11,12,13]. Inhalation of droplet nuclei containing the organism into the alveoli of a new host triggers tuberculosis infection [11,12]. The subsequent course of infection is determined by *M. tuberculosis* virulence and the host immune response [5]. Outcomes of tuberculosis infection fall into a spectrum of pathogen elimination, latent tuberculosis infection (LTBI), subclinical tuberculosis disease and clinical tuberculosis disease. The mechanisms involved in the disease development are complex. A substantial amount of work has investigated how pathogen and host factors, including local immune responses and disease tolerance, can explain the pathogenesis and susceptibility for developing tuberculosis disease [13]. Whereas the majority of the general population infected with *M. tuberculosis* do not develop active tuberculosis, PLWH are at a nearly 20-fold increased risk for reactivation of LTBI [9,14]. The risk of developing active tuberculosis after an infectious contact in immunocompetent adults is estimated to be 5%–10% during their lifetime, but this risk is increased to 5%–15% annually in PLWH [15,16].

The risk factors that contribute to developing tuberculosis disease are diverse and complicated. According to a recent research study from Switzerland, the risk factors for active tuberculosis in PLWH mainly include a plasma HIV-RNA load ≥ 1000 copies/mL, CD4 counts < 500 cells/μL, positive LTBI test (TST or IGRA), BMI < 18.5 kg/m^2^, non-white ethnicity and individual origin from a high tuberculosis epidemic region [17]. In contrast, preventive treatment of tuberculosis is a protective factor against incident tuberculosis [17]. Another study from the UK also showed that having a country of birth with a high tuberculosis burden was a significant risk factor for developing active HIV-related tuberculosis [18]. A meta-analysis among PLWH in Sub-Saharan Africa revealed that being underweight, having a low CD4 count, male gender, advanced WHO clinical stages, anemia, bedridden/ambulatory functional status, lack of isoniazid preventive therapy and lack of cotrimoxazole could boost the risk of tuberculosis occurrence [19].

Among all risk factors mentioned above, immune system changes are the most concerning and intriguing. As described in reports in the literature, a significant drop in CD4 cells in the blood and alveolus is an important contributor to the increased risk of developing active tuberculosis [17,20,21]. However, the susceptibility to tuberculosis disease increases in the early years after HIV infection, far before decreases in CD4 cells below 500 cells/μL [22], which reveals that the mechanisms of increased susceptibility go beyond the depletion of CD4 cells. There may be other alterations in the immune system induced by the HIV infection that would promote a likelihood of developing active tuberculosis. For instance, among PLWH, CD8^+^ T-cell function is disrupted [23,24]; there are decreases in apoptosis and increases in necrosis of tuberculosis-infected macrophages induced by HIV infection [25,26]. Up-regulation of *M. tuberculosis* entry receptors on macrophages in PLWH [27] have also been demonstrated to play a role in LTBI reactivation. 

## 3. When to Initiate ART in Patients with HIV and Tuberculosis Coinfection

As HIV-associated tuberculosis is very common, with a high mortality rate, ART and anti-tuberculosis treatment are both pivotal, life-saving interventions. Anti-tuberculosis therapy should be initiated immediately to control disease progression and reduce the transmission risk of tuberculosis in social surroundings. Nonetheless, ART initiation early after tuberculosis treatment has several potential risks, mainly including a high pill burden, drug toxicity, drug–drug interactions [28] and IRIS [29,30]. When to initiate ART, or rather the optimal timing of ART, in people with HIV-related tuberculosis has been universally debated and was controversial for many years. 

Since 2017, the WHO has recommended that rapid ART initiation should be offered to all PLWH following a confirmed HIV diagnosis with clinical assessment (within a week) and on the same day to people who are ready to start [31]. Likewise, ART should not be delayed too long in persons with tuberculosis but without prior ART exposure. In the SAPiT-1 study, initiation of ART during tuberculosis treatment in patients with CD4 counts < 500 cells/μL reduced mortality by 56% compared with starting ART after tuberculosis treatment completion [32]. A systematic review and meta-analysis of six randomized controlled trials enrolling 2272 participants found that compared to delayed ART initiation (eight weeks to twelve weeks after anti-tuberculosis treatment starting), early ART initiation (within four weeks after anti-tuberculosis treatment starting) could significantly reduce all-cause mortality in patients coinfected with HIV and tuberculosis despite an increased risk for immune reconstitution disease [33]. Another meta-analysis of eight randomized controlled trials including 4568 participants that evaluated early versus delayed ART initiation (1–4 weeks vs. 8–12 weeks after initiation of tuberculosis treatment) or deferred ART initiation (after the end of tuberculosis treatment) showed that early ART initiation in PLWH with newly diagnosed pulmonary tuberculosis would improve survival only in those with CD4 counts less than 50 cells/μL, while a mortality benefit from early ART was not found among those with CD4 counts more than 50 cells/μL [34]. Recently, Burke et al. published a systemic review and meta-analysis with nine trials conducted between 2004 and 2014, finding that among people with all CD4 counts, there was no difference in mortality with earlier ART (≤4 weeks) versus later ART (>4 weeks) [29]; however, there was lower risk of death in the earlier ART group among people with CD4 counts ≤ 50 cells/μL. Based on much of the evidence for rapid ART, in March of 2021, the WHO recommended that ART should be started within two weeks of tuberculosis treatment at any CD4 count [35], especially for those with CD4 ≤ 50 cells/μL. Three high HIV burden countries in Africa (Malawi, Zambia and Uganda) have already issued national guidelines recommending ART within or at two weeks after starting tuberculosis treatment for people with tuberculosis and HIV at all CD4 cell counts [36,37,38].

Tuberculous meningitis (TBM) is a severe central nervous system (CNS) infection. There is a high proportion of TBM and HIV co-infection. As shown in one study, the prevalence of HIV infection is 30% in patients with confirmed TBM, and 12.1% in patients with suspected TBM [39]. The diagnosis of TBM faces challenges. There are no FDA-approved/cleared diagnostic tests for the detection of *M. tuberculosis* from cerebrospinal fluid (CSF) to aid in the diagnosis of TBM, though there is an expanding toolbox for detecting neurotropic organisms from CSF [40]. The possibilities of death for HIV/AIDS patients with TBM are comparatively high. Profound immunodeficiency, late presentation, advanced CNS disease, a high risk of severe immune reconstitution with rapid ART initiation and high rates of comorbidity all conspire towards a dismal prognosis [41]. In March 2021, WHO recommendations of ART initiation within two weeks of tuberculosis treatment at any CD4 count were mainly targeted to individuals with extracranial tuberculosis. However, the optimal time to initiate ART in patients with HIV-related TBM remains uncertain and may differ from extracranial tuberculosis. One randomized, double-blind, placebo-controlled trial compared immediate ART (initiated within 7 days of commencing tuberculosis treatment) with deferred ART (initiated after 2 months of tuberculosis treatment) in patients with HIV-associated TBM, with the average CD4 count in two groups less than 50 cells/µL [42]. Here, immediate ART could not reduce the risk of 9-month death. It seems that the timing of ART initiation makes no appreciable difference to survival probability. Based on this study, the Official American Thoracic Society/Centers for Disease Control and Prevention/Infectious Diseases Society of America recommended that patients with TBM should not start ART within eight weeks after anti-tuberculosis treatment initiation, regardless of CD4 count [43]. Nevertheless, many experts recommend that ART in patients with HIV-associated TBM should be started within the first 2–8 weeks after initiating anti-tuberculosis treatment. In 2023, the Europe AIDS Clinical Society recommended guidelines for initiating ART in patients with TBM as follows [44]: in persons with CD4 counts < 50 cells/µL, ART should be initiated within the first 2 weeks after initiation of tuberculosis treatment, while ART initiation should be delayed for 4 weeks in all other cases. The WHO guidelines of 2021 recommend that ART should be delayed at least four weeks (and initiated within eight weeks) after anti-TBM treatment was initiated and corticosteroids should be considered as adjuvant treatment [35,45], which is consistent with the guideline recommendations in China [46]. More details on the timing of starting ART in people coinfected with HIV and tuberculosis from international mainstream guidelines are listed in Table 1 [35,44,45,46,47,48,49].

In addition, to two other special groups of PLWH—persons with drug-resistant tuberculosis and pregnant women with tuberculosis—more attention should be paid to the initiation timing of ART. There is a high proportion of drug-resistant tuberculosis in PLWH. Although randomized clinical study data to guide the optimal timing for ART initiation are lacking [47], the WHO recommends initiating ART for all patients with HIV and drug-resistant tuberculosis requiring second-line anti-tuberculosis drugs, irrespective of CD4 cell count, as early as possible (within the first 8 weeks) following the initiation of tuberculosis treatment [50]. Pregnant women with active tuberculosis and HIV infection should start ART immediately, regardless of CD4 cell count and tuberculosis sites, in order to control the virus rapidly and reduce HIV transmission to newborns [47].

## 4. Drug and Drug Interactions between Anti-Tuberculosis Drugs and ARV Agents

Pharmacokinetic (PK) interactions frequently occur between ARV drugs and concurrently administered medications, resulting in potential alterations in drug exposure, either heightened or diminished [47]. The most significant interactions are between rifamycins (rifampin, rifabutin and rifapentine) and ARV drugs. Rifampin is a potent inducer of hepatic cytochrome P450 (CYP)—mostly 3A and 2C subfamilies accounting for more than 80% of the CYP isoenzymes [51,52], P-glycoprotein and uridine diphosphate glucuronosyltransferase (UGT) 1A1 enzymes, while rifabutin and rifapentine are CYP3A substrates and inducers [47]. All rifamycins would cause a certain degree of reduction in ARV drug exposure when used concomitantly. Despite the potential drug interactions with ARV agents, rifamycins, as the cornerstone of anti-tuberculosis therapy, are irreplaceable in drug-susceptible tuberculosis treatment owing to their sterilizing ability [28,47]. One research study has reported that rifamycin-lacking regimens have significantly higher failure/relapse rates (11.7% vs. 6%, *p* = 0.002) [53]. The relative potency of rifamycins as CYP3A inducers is rifampin = rifapentine > rifabutin [54]. The ARV drugs which are most affected by rifamycins include all the protease inhibitors (PIs), the non-nucleoside transcriptase inhibitors, the integrase strand transfer inhibitors (INSTIs), the gp-120-attachment inhibitor fostemsavir and the CCR5 antagonist maraviroc, whereas most nucleos(t)ide reverse transcriptase inhibitors, the fusion inhibitor enfuvirtide and the CD4 post-attachment inhibitor ibalizumab are not anticipated to exhibit noteworthy drug interactions with rifamycins [47]. Below, PK interactions between rifamycins and ARV drugs are introduced briefly based on several updated guidelines [44,47,48] and more details are listed in Table 2.

### 4.1. When Used with Rifampin Concomitantly

All PIs, doravirine, etravirine, nevirapine, rilpivirine, bictegravir, cabotegravir and elvitegravir/cobicistat, are not recommended when used with rifampin concomitantly. Tenofovir alafenamide (TAF), a new kind of nucleos(t)ide backbone drug, is a substrate of the drug transporter P-glycoprotein and may have more potential interaction with rifampin than tenofovir disoproxil fumarate (TDF). However, a study conducted in HIV-negative healthy volunteers showed that concentrations of intracellular tenofovir-diphosphate, the active form of tenofovir, were higher in those with TAF/emtricitabine plus rifampin than in those with TDF only [55]. It should be noted that the concurrent use of TAF and rifampin has not been studied in patients taking full-dose ARV agents and anti-tuberculosis regimens to verify PK and antiviral efficacy. In a recent observational study of serial cases, Mohzari et al. assessed the viral control effectiveness of the concomitant use of rifampin and TAF/emtricitabine once daily in PLWH and showed that all seven patients had a suppressed viral load after completing six months anti-tuberculosis therapy whether the HIV viral loads at baseline were detectable or not [56]. The results indicated that the dose of TAF may not require adjustment when co-administrated with rifampin in PLWH. Randomized clinical trials with larger samples conducted in PLWH are needed to validate this conclusion. 

Dolutegravir (DTG), raltegravir (RAL) and maraviroc dosages need to be adjusted when used with rifampin. The recommended daily dose of DTG is increased from 50 mg once daily to 50 mg twice daily if used together with a rifampin-based anti-tuberculosis therapy. RAL 800 mg twice daily (standard dose 400 mg twice daily) is recommended when used with rifampin. Maraviroc 600 mg twice daily (standard dose 300 mg twice daily) is recommended in combination with rifampin, but alternative ARV or antimycobacterial drugs should be considered to replace maraviroc if used together with a strong CYP3A inhibitor.

### 4.2. When Used with Rifabutin Concomitantly

The ARV agents including PIs with cobicistat, rilpivirine (intramuscular), TAF, bictegravir, cabotegravir (intramuscular) and elvitegravir/cobicistat are not recommended to use together with a rifabutin-based anti-tuberculosis regimen. The daily dose of doravirine and rilpivirine (oral) needs to be increased when used together with rifabutin. The dose of doravirine should be increased to 100 mg twice daily (standard dose 100 mg once daily), and the dose of oral rilpivirine should be increased from 25 mg once daily to 50 mg once daily.

The dose of DTG does not need to increase to 50 mg twice daily (standard dose 50 mg once daily) theoretically when used with rifabutin concomitantly due to rifabutin’s weaker potential as a UGT1A1 inducer than rifampin. However, our previous study in a real setting discovered that maximum and trough concentrations of DTG (50 mg once daily) with rifabutin were both significantly lower than that of DTG used in HIV/AIDS patients without tuberculosis. Hence, DTG 50 mg once daily together with rifabutin may only serve as an alternative option for HIV and tuberculosis co-infected patients [57].

## 5. Preferred ART for People with HIV-Related Tuberculosis

An ART regimen containing efavirenz plus two nucleos(t)ide analogues, including TDF, lamivudine, zidovudine, abacavir and emtricitabine, combined with a rifamycins-containing anti-tuberculosis treatment is the preferred co-treatment regimen for HIV-associated tuberculosis [48]. A rifampin-based antitubercular therapy and an efavirenz-based ARV regimen are the first choice [58]. Importantly, although rifampin induces CYP isoenzymes, potentially lowering efavirenz exposure, several studies have demonstrated that efavirenz of a standard 600 mg per day dose is suitable for most PLWH without dose adjustment, even for patients with a higher body weight [59,60,61]. Interpretations of these results are that about 20% of a population of African, Thai or Indian ancestry are low metabolizers (usually with the TT genotype of CYP2B6 G516T polymorphism) of efavirenz and thus have high plasma efavirenz concentrations. In addition, isoniazid, as the inhibitor of the CYP2A6 enzyme (an accessory metabolic pathway for efavirenz), further increases blood efavirenz concentrations [62]. According to WHO guideline, efavirenz at a low dose (400 mg per day) combined with two nucleos(t)ide analogues is recommended as the alternative first-line therapy for PLWH without tuberculosis [45]. Some researchers considered that low dose efavirenz might be safe in patients with HIV-associated tuberculosis when co-administrated with rifampin. A study, enrolling 26 participants coinfected with HIV and tuberculosis, found that there was only a limited reduction (<25%) in efavirenz area under curve when efavirenz 400 mg per day was co-administrated with rifampin, and efavirenz blood concentration was sufficient to maintain virologic suppression [63]. However, due to the restricted cases in this trial, the guidelines in the United States continue to recommend a standard dose of efavirenz 600 mg once daily for individuals receiving rifampin therapy [47,48].

Alternatives to efavirenz for patients with HIV-*M. tuberculosis* coinfection are PIs or INSTIs. The plasma concentrations of PIs would decrease by >75% when co-administered with rifampin [58], which may be overcome by doubling the dose of PIs or through the strategy of increasing the dose of boosters—ritonavir or cobicistat; however, high rates of hepatotoxicity have been reported in healthy volunteers [64,65,66]. Nonetheless, a recent trial in people with HIV-associated tuberculosis discovered that the use of double-dose lopinavir/ritonavir in combination with rifampin yielded satisfactory safety, drug concentration and response to tuberculosis treatment [67]. Certain experts may view this as a viable option when a PIs-based ART regimen is required during tuberculosis treatment [48]. Further high-quality research on the safety and efficacy of PIs is needed for HIV-*M. tuberculosis*-coinfected people with rifampin treatment. INSTIs are mainly metabolized by UGT1A1 enzymes; the activity and expression of which would be induced by rifampin but less so by rifabutin. RAL and DTG are the only two INSTIs to be used with a rifampin-containing antituberculosis regimen, but not bictegravir, cabotegravir or elvitegravir/cobicistat [47]. Theoretically, RAL used together with rifampin will reduce the concentrations of RAL markedly and increasing the RAL dose to 800 mg twice daily could alleviate the interaction of PK [68]. A multicenter, phase 2, randomized trial in 2014 disclosed that RAL 400 mg twice daily had similar virological suppression compared with efavirenz 600 mg once daily or RAL 800 mg twice daily plus TDF and lamivudine after completing a 24-week follow-up, suggesting that it may be unnecessary to double the RAL dose in patients co-infected with HIV and tuberculosis [69]. However, another randomized phase 3 trial in 2021 demonstrated that although 61% of patients in the standard dose RAL (400 mg twice daily) group vs. 66% in the efavirenz (600 mg per day) group had realized virological suppression, non-inferiority of RAL compared with efavirenz was not shown at 48 weeks follow-up in individuals coinfected with HIV and tuberculosis treated with rifampin-containing therapy [70]. Thus, standard dose RAL, in spite of good tolerance, serves as an alternative option in selected patients with tuberculosis [70]. 

DTG, a new generation of INSTIs, is widely prescribed due to its good tolerability, potent antiviral activity and high genetic barrier to resistance. The area under the curve or trough concentrations of DTG might sharply reduce when prescribed with rifampin, which has been demonstrated not only among healthy volunteers [71,72] but also in patients coinfected with HIV and tuberculosis [73]. But the reduction could be partially compensated by doubling the DTG dosage to 50 mg twice daily or even 100 mg twice daily [72,73]. In a multi-center, randomized trial, comparing DTG 50mg twice daily with efavirenz 600 mg daily when co-administrated with rifampin, the response rate at week 48 was 75% vs. 82%, and there were no deaths in either group and no DTG-related acquired resistance [74]. DTG 50 mg twice daily is comparable to efavirenz in efficacy and safety and thus can be used as an alternative to efavirenz in PLWH with rifampin-based tuberculosis treatment [74]. However, some recent studies proposed that 50 mg twice daily DTG with rifampin might be unnecessary. A retrospective study from Botswana recruited 739 patients with HIV-associated tuberculosis concomitantly administered rifampin- and DTG-based regimens, and found that 44% patients with DTG once daily dosing realized similar viral suppression to 53% of patients who were prescribed DTG twice daily dosing (adjusted *p* = 0.039) [75]. Furthermore, Rulan Griesel and colleagues conducted a phase 2 randomized, double-blind and placebo-controlled trial in Khayelitsha, South Africa (in total, 108 participants with HIV-associated tuberculosis were treated with a rifampin-based regimen), finding similar viral suppression rates in groups of DTG 50 mg once versus twice daily [76]. Despite inadequate plasma trough concentrations, no treatment-emergent DTG-resistant mutations were detected up to week 48 in the 19 participants with study-defined virological failure [76]. The results are encouraging and might offer a cautious level of assurance to clinicians and public health authorities globally regarding the efficacy and safety of a once daily DTG regimen [77]. These findings should be carefully generalized to all patients cotreated with HIV and tuberculosis on account of only the phase 2 trial with a small sample size and non-comparative design [78]. Moreover, including patients only with CD4 counts > 100 cells/μL and excluding patients with previous HIV treatment failure would also limit generalizability of the study conclusion. Interestingly, in light of the extensive literature reporting the positive association between race/ethnicity and plasma concentrations of ARV agents in PLWH, such as efavirenz and TDF, Dario Cattaneo speculated that due to the genetic heterogeneity of DTG metabolism, African patients treated with DTG 50 mg once daily with rifampin might be exposed to higher drug concentrations of DTG compared with Caucasian cohorts using the same DTG dose with rifampin [77], a hypothesis that requires further study. The optimal dose of DTG in patients with HIV-related tuberculosis concomitantly used with rifampin is still a work in progress and not conclusive, and a phase 3 trial may be required to offer evidence for revising guidelines and policies [78]. Up to now, most guidelines from different countries continue to recommend DTG at a dose of 50 mg twice daily when used together with a rifampin-containing tuberculosis regimen [45,46,48]. 

## 6. The Development of TB-IRIS after ART Initiation

TB-IRIS is a clinical condition caused by ART-induced restoration of immune responses directed against *M. tuberculosis*. IRIS has been linked to various pathogens and autoimmune diseases, but mycobacterial infections are the most common cause of IRIS. It presents within the first 2 months of ART, usually in the first 2–3 weeks. There was no consensus on a case definition and diagnosis standard for IRIS until the international network for the study of HIV-associated IRIS was formed in 2006, which has accelerated the research pace and clinical management of IRIS. 

### 6.1. Definition, Diagnosis and Epidemiology of IRIS

The criteria outlined in these case definitions include a confirmed HIV diagnosis, a temporal link with the initiation of ART, verification of ART responsiveness, clinical deterioration involving an inflammatory process and the exclusion of alternative causes accounting for the deterioration [79]. TB-IRIS mainly includes two forms: paradoxical TB-IRIS (the deterioration of a treated infection) and unmasking TB-IRIS (a new presentation of previously occult or subclinical infection). For paradoxical TB-IRIS, patients have been diagnosed with active tuberculosis pre-ART; within the first few weeks and up to 3 months after initiating ART, patients undergo new, recurring or exacerbated features of tuberculosis, which may encompass lymph node swelling and abscess formation, serositis and radiographic deterioration [79]. A diagnosis of unmasking TB-IRIS may be made as follows: a lack of evidence for tuberculosis infection, not receiving anti-tuberculosis treatment when ART is initiated and then presentation with active tuberculosis within 3 months of starting ART. Unmasking TB-IRIS usually presents with notable clinical manifestations like lymphadenitis, abscess and respiratory failure [79]. However, owing to ambiguous knowledge and definition, as well as the lack of definitive biomarkers, diagnosis of unmasking TB-IRIS remains a challenge. It is still difficult to differentiate unmasking TB-IRIS from the presentation of non-IRIS tuberculosis infection following ART [80]. 

The incidence rates of TB-IRIS vary from different countries and regions, fluctuating widely, accompanied with a certain mortality. In 2020, a retrospective study from Beijing, China, found that 22.6% (45/199) of HIV and tuberculosis co-infected patients developed paradoxical TB-IRIS after ART initiation [81], revealing a comparatively high incidence rate of paradoxical TB-IRIS. A meta-analysis of 40 studies found that the overall incidence of paradoxical TB-IRIS was 18% among individuals with HIV-associated tuberculosis, with mortality attributable to TB-IRIS of 2% [82]. Unmasking TB-IRIS was observed in 1–4% of patients starting ART [80,83]. Patients with TBM, however, were at high risk of developing IRIS (47%), with a mortality rate of up to 30% following development of IRIS [84,85,86]. 

### 6.2. Risk Factors and Biomarkers of IRIS

Although the underlying mechanisms of TB-IRIS pathogenesis have not been completely elucidated, a few factors or biomarkers may be predictive for TB-IRIS development. 

#### 6.2.1. Clinical Risk Factors

Initiating ART early is an essential IRIS risk factor shown in prior studies. A study showed that the IRIS incidence was 3.76 vs. 1.53 cases per 100 person months in the earlier vs. later ART group [87]. In another large multi-center trial, the IRIS rates were 11% vs. 5% in the immediate ART group vs. early ART group [88]. A study from Narendran even found that a short interval between starting antitubercular therapy and ART of less than 30 days increased TB-IRIS risk 20.2 times [89]. Patients with a low CD4 count at the time of ART initiation are more likely to develop TB-IRIS. According to findings from the SAPiT trial, the TB-IRIS incidence increased prominently following a decline in CD4 count, which was 5.6, 12.3 and 23.1 per 100 person years with CD4 counts > 200 cells/μL, between 50 and 200 cells/μL and <50 cells/μL, respectively [90]. However, these researchers cautioned that the high IRIS risk from early ART initiation must be balanced with the increased survival benefit [90] previously shown in populations with CD4 counts < 50 cells/μL. Apart from early ART initiation and low CD4 count, a high viral load [90,91], the degree of immunological restoration after ART treatment [92,93], disseminated tuberculosis [93], Black ethnic origin [94], positive urine lipoarabinomannan antigen [95], elevated extrapulmonary extracellular matrix turnover [96] and plasma hemoglobin less than 90 g/L at baseline [89] are also regarded as predictive factors of TB-IRIS in numerous studies.

#### 6.2.2. Immune and Metabolic Biomarkers

As TB-IRIS results from excessive inflammatory reactions against tuberculosis antigens, cytokines and acute-phase reactants from ART-induced restoration of the immune system are likely to forecast the development of TB-IRIS. A study from Narendran discovered that the combination of IL-6 and CRP assays contributed to the sensitivity for IRIS detection 12% higher than IL-6 and 16% higher than CRP alone (sensitivities: IL-6 + CRP = 92%) [89]. Additionally, other studies manifested that pre-ART levels of IFN-λ, sCD14, CCL-2, CXCL-10 and IL-18 were independently associated with the IRIS risk [97,98,99,100]. A recent metabolomic profiling study generated eight biomarker predictors, including components of arachidonic acid, linoleic acid and glycerophospholipid metabolism, which could distinguish TB-IRIS from non-IRIS with a sensitivity of 92% and specificity of 100% [101].

#### 6.2.3. Genetic Biomarkers

Several studies have demonstrated that genetic biomarkers have the potential to forecast the risk of IRIS development or severity. A higher incidence of severe IRIS among patients with mutant LTA4H genotypes (CT and TT) was observed compared to those with the wild type (CC) [102]. Another study uncovered a genetic association of HLA-B*41, KIR2DS2 and KIR + HLA-C pairs with IRIS onset among tuberculosis and HIV co-infected individuals [103]. Furthermore, the polymorphisms in the IL-6, IL-18, TNF-α, vitamin D receptor and natural resistance-associated macrophage protein 1 genes were associated with differential risks for *M. tuberculosis* and *Mycobacterium avium complex* IRIS in two studies [104]. Although a variety of gene markers were detected in association with IRIS, the proportions of genotypes in different ethnicities vary remarkably and need further exploration. 

### 6.3. Prevention and Treatment of IRIS

Most mild to moderate TB-IRIS is not life-threatening, even if it is self-limiting without specific intervention. However, substantial and marked inflammation reaction may cause death especially in cases with TBM-associated IRIS, and therefore anti-inflammatory therapy is appropriate. 

Corticosteroids are a first-line therapy to inhibit excessive inflammation response to tuberculosis antigens in IRIS. A randomized placebo-controlled trial discovered that patients clinically diagnosed with paradoxical TB-IRIS (mainly mild TB-IRIS) who received a 4-week course of prednisone had a reduced need for hospitalization, quicker resolution of symptoms and a better quality of life without excess severe adverse events, compared to those who received a placebo [105,106]. Although corticosteroids were previously demonstrated to prevent IRIS in patients with non-neurological tuberculosis and HIV [107], a recent study found that preventive use of dexamethasone did not reduce IRIS incidence among patients with HIV-associated TBM [108]. To sum up, in patients with HIV and tuberculosis coinfection, corticosteroids are helpful to prevent and treat non-life-threatening TB-IRIS but not to prevent the TBM-related IRIS incidence. Whether it is effective to treat TBM-associated IRIS is unclear and in need of further investigation.

Apart from corticosteroids, other anti-inflammatory agents have also been used to treat TB-IRIS, including NSAIDs, immunomodulatory drugs (thalidomide, TNF-α inhibitors such as infliximab and adalimumab), leukotriene antagonists (e.g., montelukast and zileuton), pentoxifylline, hydroxychloroquine, etc. Some of them may be effective for treating cases of steroid-refractory TB-IRIS or even intractable intracranial tuberculosis-associated IRIS. However, there is no adequate evidence to recommend any of these agents for routine use and further rigorous evaluation in randomized trials is required.

## 7. Conclusions

Tuberculosis remains a prominent opportunistic infection and the leading cause of death in people with HIV. The complex relationship between HIV and *M. tuberculosis* challenges effective treatment with ART for those co-infected patients. Early initiation of ART is critical to improve the prognosis. For non-TBM patients, it is recommended to start ART as early as possible, usually within 2 weeks of initiating antituberculosis treatment. Given interactions between ARV drugs and rifamycins, the ART regimen is unique and must be carefully assessed. INSTIs, especially DTG, have become one of the main choices for ART in co-infected individuals. Preventive use of corticosteroids may have a certain effect on reducing TB-IRIS, but it is necessary to have a good understanding of the target population and further research is required to gather more evidence. In the future, more research is needed to continuously optimize personalized ART strategies and confirm the optimal timing of ART initiation among patients with special situations, such as TBM. We hope, with close collaboration and advances in scientific understanding, the challenges and obstacles of ART for patients with HIV-associated tuberculosis will be conquered gradually, eventually bringing more favorable prognoses.

## Figures and Tables

**Table 1 viruses-16-00494-t001:** ART initiation timing recommendations from mainstream guidelines for patients with HIV-associated tuberculosis.

Source of Guideline	Population or Clinical Status	Timing of ART Initiation
WHO [35,45]	– PLWH with suspected TB symptoms (except for signs and symptoms of meningitis)	– Rapid ART initiation while investigating for TB, with close follow-up within 7 days to start TB treatment if TB is confirmed
	– Being treated for active TB except of TBM	– Within 2 weeks after TB treatment, regardless of CD4 count
	– Being treated for TBM	– 4–8 weeks after treatment for TBM
DHHS [47]/IDSA [48]	– Active TB except of TBM	
	• CD4 counts < 50 cells/μL	• Within 2 weeks after TB treatment
	• CD4 counts ≥ 50 cells/μL	• Within 8 weeks after TB treatment
	• During pregnancy	• As early as feasible, regardless of CD4 count
	– TBM	– If starting ART early, monitoring adverse events and deaths closely
EACS [44]	– Active TB except of TBM	– Within 2 weeks after TB treatment
	– TBM	
	• CD4 counts < 50 cells/μL	• Within the first 2 weeks after TB treatment, while close monitoring
	• CD4 counts ≥ 50 cells/μL	• Delayed for 4 weeks after TB treatment
IAS-USA [49]	– Active TB without evidence of TBM	– Within 2 weeks after TB treatment, especially for those with CD4 count < 50 cells/μL
	– TBM	– Within 2 weeks after TB treatment and co-administering high-dose steroids
CMA [46]	– Active TB except of TBM	– Within 2 weeks after TB treatment
	– TBM	– 4 to 8 weeks after TB treatment

ART = antiretroviral therapy; HIV = human immunodeficiency virus; PLWH = people living with HIV; TB = tuberculous; TBM = tuberculous meningitis; WHO = World Health Organization; DHHS = Department of Health and Human Services; IDSA = Infectious Diseases Society of America; EACS = European AIDS Clinical Society; IAS-USA = International Antiviral Society-USA Panel; CMA = Chinese Medical Association.

**Table 2 viruses-16-00494-t002:** Pharmacokinetic interactions between anti-tuberculosis drugs and ARV agents.

Anti-TB Drugs	ARV Agents	PK Interaction	Clinical Comments and Recommended Doses
Rifampin	All PIs, DOR, ETR, NVP, RPV (IM and PO), CAB (IM and PO), BIC and EVG/c	– ARVs concentrations ↓ > 50%.	– Not recommended
	EFV	– AUC of EFV ↓ 26%	– EFV 600 mg once daily (but not 400 mg once daily) is recommended to use with rifampin; monitoring virologic response is needed.
	TAF	– Compared with TDF alone, TAF plus rifampin, AUC of TFV-DP ↑ 4.2-fold; – Compared with TAF alone, TAF plus rifampin, AUC of TAF ↓ 55%, AUC of TFV-DP ↓ 36%; – Compared with TAF 25 mg once daily alone, TAF 25 mg twice daily plus rifampin, AUC of TAF ↓ 14%, AUC of TFV-DP ↓ 24%.	– Avoid concurrent administration unless benefits outweigh risks. If used concomitantly, closely monitor antiviral efficacy.
	DTG	– Compared to DTG 50 mg twice daily, rifampin with DTG 50 mg twice daily, AUC of DTG ↓ 54%, C_min_ ↓ 72%; – Compared to DTG 50 mg once daily, rifampin with DTG 50 mg twice daily, AUC of DTG ↑ 33% and C_min_ ↑ 22%.	– DTG 50 mg twice daily for patients is recommended if without suspected or confirmed INSTI-related resistance mutations. Use rifabutin instead of rifampin if with INSTI-related mutations.
	RAL	– Compared to RAL 400 mg twice daily, RAL 400 mg per day, AUC of RAL ↓ 40% and C_min_ ↓ 61%; – Compared to RAL 400 mg twice daily, rifampin with RAL 800 mg twice daily, AUC of RAL ↑ 27% and C_min_ ↓ 53%.	– RAL 800 mg twice daily is recommended with monitoring antiviral efficacy closely or replace rifampin with rifabutin. RAL 1200 mg once daily plus rifampin is not recommended.
	MVC *	– AUC of MVC ↓ 63%.	– Increase MVC dose to 600 mg twice daily.
	TDF	– No significant influence	– No dose adjustment needed
Rifabutin	PIs/c, RPV(IM), TAF, BIC, CAB (IM), and EVG/c	– Boold concentrations of those ARV agents decrease significantly	– Not recommended
	PIs/r	– Concentrations of rifabutin or its metabolite would increase significantly	
	DOR	– AUC of DOR ↓ 50%	– Increase DOR dose to 100 mg twice per day. There is no need to adjust rifabutin dose.
	EFV	– Rifabutin ↓ 38%	– The suggested rifabutin dose range is 450–600 mg per day.
	RPV (PO)	– Compared to RPV 25 mg once daily, the co-administration of RPV 50 mg once daily with rifabutin, AUC and C_min_ of RPV ↔	– The RPV dose is increased to 50 mg once daily. No dose adjustment is needed for rifabutin.
	ETR	– AUC of rifabutin and metabolite ↔; AUC of ETR ↓ 37%	– Rifabutin is not suggested to co-administer with ETR plus PIs/r. Without PIs/r, rifabutin 300 mg once daily plus ETR is suitable.
	MVC	– AUC of MVC ↔ and C_min_ ↓ 30%	– MCV 150 mg twice daily is recommended when used with a strong CYP3A inhibitor, while 300 mg twice daily is suggested if not.
	NVP, TDF, DTG, RAL, and CAB (PO)	– No significant influence	– No dose adjustment needed
Rifapentine	All PIs, DOR, ETR, NVP, RPV (IM and PO), TAF, BIC, EVG/c, CAB (IM and PO), and MVC	– Concentrations of those ARVs decrease significantly	– Not recommended
	DTG	– AUC of DTG ↓ 26% and C_min_ ↓ 47% when co-administrated with rifapentine 900 mg once weekly	– DTG 50 mg per day may be suggested to co-administer with once weekly rifapentine (but not once daily rifapentine) in individuals with viral suppression, and it is needed to monitor antiviral efficacy.
	RAL	– AUC of RAL ↑ 71% and C_min_ ↓ 12% when co-administrated with rifapentine 900 mg once weekly; – C_min_ of RAL ↓ 41% when co-administrated with rifapentine 600 mg once daily.	– RAL 400 mg twice daily is recommended with once weekly rifapentine (but not once daily rifapentine) without dose adjustment.
	EFV and TDF	– No significant influence	– No dose adjustment needed

ARV = antiretroviral; AUC = area under curve; BIC = bictegravir; c = cobicistat; CAB = cabotegravir; C_min_ = minimum drug concentration; CYP3A = cytochrome P450 3A; DOR = doravirine; DTG = dolutegravir; EFV = efavirenz; ETR = etravirine; EVG/c = elvitegravir/cobicistat; IM = intramuscular; INSTI = integrase strand transfer inhibitors; MVC = maraviroc; NVP = nevirapine; PIs = protease inhibitors; PIs/c = protease inhibitors/cobicistat; PIs/r = protease inhibitors/ritonavir; PO = oral; r = ritonavir; RAL = raltegravir; RPV = rilpivirine; TAF = tenofovir alafenamide; TB = tuberculosis; TDF = tenofovir disoproxil fumarate; TFV = tenofovir; TFV-DP = tenofovir diphosphate; ↓ means decrease; ↑ means increase; ↔ means no change. * Maraviroc needs to be substituted to alternative ARV or antimycobacterial if used concomitantly with a strong CYP3A inhibitor.

## Data Availability

No new data were generated in this work.

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
