# Peer review of "Advances in Antiretroviral Therapy for Patients with Human Immunodeficiency Virus-Associated Tuberculosis"

_viruses, 2024, doi:10.3390/v16040494_

Round 1

Reviewer 1 Report

Comments and Suggestions for Authors

Lines 55-57 and 63-64: Cited about China. It is not the objective of this paper, the jump from Globally to locally (China) cannot be done unless there are more examples (different Countries). 

Line 118: ART should not be delayed in persons with opportunistic infections: It should be in case of Cryptococcal meningitis. 

Line 175: Maybe an introduction for this paragraph as it is about two different tuberculosis risks groups.

Table 2 (page 7): PK interactions: DTG: not clearly understood (twice daily or once daily alone).

Line 164 (page 12): urine. Should be "positive urine" to be better understood. 

Comments on the Quality of English Language

Line 84: nicity, individual. Maybe change "," for "and".

Line 89: Bedridden OR ambulatory? or both of them? 

Line 14 (page 6): "has been reported": not needed. 

Lines 90-91 (page 10): that - that is repeated. 

Line 103 (page 10): phage should be phase. 

Lines 224-225 (page 13): "more research is needed" is written twice, look for a different option. 

Author Response

Response to Reviewers' Comments:

Reviewer 1

  1. Comments and Suggestions for Authors

Comment 1: Lines 55-57 and 63-64: Cited about China. It is not the objective of this paper, the jump from Globally to locally (China) cannot be done unless there are more examples (different Countries).

Response:We thank the reviewer for this constructive and reasonable suggestions. We have deleted the sentences about epidemiology of HIV-related tuberculosis in China.

Comment 2: Line 118: ART should not be delayed in persons with opportunistic infections: It should be in case of Cryptococcal meningitis. 

Response: We are grateful for your valuable comments. Actually, ART should be initiated immediately in patients with cryptococcal meningitis without any delay, but not including those with tuberculosis infection even tuberculous meningitis. The previous expression in our manuscript was not rigorous and exact enough. We have substituted the statement with the following sentence “not be delayed too long in persons with tuberculosis, without prior ART exposure”.

Comment 3: Line 175: Maybe an introduction for this paragraph as it is about two different tuberculosis risks groups.

Response: We thank the reviewer for this enlightening advice for our study. We have added an introduction for this paragraph with the sentence “Furthermore, for two other special groups in PLWH— persons with drug-resistance tuberculosis and the pregnant women infected with tuberculosis, much attention should be paid to initiation timing of ART”. We hope it suitable to plays a bridging role between the preceding and the following. Hopefully, the modification is eligible to be accepted.

Comment 4: Table 2 (page 7): PK interactions: DTG: not clearly understood (twice daily or once daily alone).

 Response: Thank you for the suggestion. We are so sorry for the former expression with ambiguity. The phrase “DTG 50mg twice daily alone” means DTG 50mg twice daily per day without rifampin concurrently. In order to avoid confusion, I have exchanged it with this phrase “DTG 50mg twice daily”. Correspondingly, the phrase “DTG 50mg once daily alone” has been replaced with “DTG 50mg once daily”.

Comment 5: Line 164 (page 12): urine. Should be "positive urine" to be better understood.  

Response: We are very appreciated for your insightful proposition. We have added “positive” before the word “urine” in the revised manuscript.

  1. Comments on the Quality of English Language

Line 84: nicity, individual. Maybe change "," for "and".

Line 89: Bedridden OR ambulatory? or both of them? 

Line 14 (page 6): "has been reported": not needed. 

Lines 90-91 (page 10): that - that is repeated. 

Line 103 (page 10): phage should be phase. 

Lines 224-225 (page 13): "more research is needed" is written twice, look for a different option. 

Response: Thank you for the useful suggestions. We have made all the modifications in the revised paper accordingly.

Reviewer 2 Report

Comments and Suggestions for Authors

Tuberculosis is one of the leading opportunistic infections and the most important cause of death in patients with human immunodeficiency virus (HIV) infection, despite near-universal access to antiretroviral therapy (ART) and tuberculosis preventive therapy. For patients with active tuberculosis but not yet receiving ART, starting ART after anti-tuberculosis treatment can complicate clinical management due to drug toxicities, drug-drug interactions, and immune reconstitution inflammatory syndrome (IRIS) events. Over the years, there has been significant progress in ART-related research for individuals who are coinfected with HIV and M. tuberculosis. 

The purpose of this review was to help doctors in treating patients who are co-infected with both HIV and tuberculosis. The authors collected and analyzed the most recent research on antiretroviral therapy (ART) for people with HIV-associated tuberculosis. The review is well-written and organized, and the tables are clear and informative, providing valuable insights into the latest treatment guidelines for patients with HIV/TB. The topics cover the major challenges in treating patients with HIV/TB.

It is recommended to have a native English speaker review the manuscript to ensure its accuracy and clarity. For instance, on page 8, item 5, line 32, the sentence reads "Importantly, although rifampin induces CYP 450 isoenzymes potentially lowing efavirenz exposure", when it should read "Importantly, although rifampin induces CYP 450 isoenzymes, potentially lowering efavirenz exposure.”

It is essential that the authors adopt a consistent terminology to describe HIV infection throughout the manuscript. This will prevent any confusion and promote clarity. They should use a single term, such as "people living with HIV/AIDS," and avoid using "HIV-positive people" or "HIV-infected individuals.”

I suggest the authors include the following sentence in their writing: "IRIS has been linked to various pathogens and autoimmune diseases, but mycobacterial infections are the most common cause of IRIS." The way it is written, it seems that IRIS is only associated with HIV/TB co-infection. Still, many opportunistic infections, such as fungal, viral, and parasitic infections, can also be linked to IRIS.

The authors discuss unmasking TB-IRIS in just four lines (135-138) in their review article. However, I believe that they should elaborate more on this definition of IRIS. For instance, they can include information about the percentage of individuals who develop this type of IRIS and the risk factors associated with the emergence of unmasked IRIS.

The authors discussed the predictive factors for IRIS in lines 163-166. However, some additional factors, such as the antigenic load, the degree of immunological restoration after cART treatment, and the host's genetic susceptibility, are also important and should be included. It's worth noting that these mechanisms can interact with each other, leading to the development of the syndrome.

Comments on the Quality of English Language

It is recommended to have a native English speaker review the manuscript to ensure its accuracy and clarity. For instance, on page 8, item 5, line 32, the sentence reads "Importantly, although rifampin induces CYP 450 isoenzymes potentially lowing efavirenz exposure", when it should read "Importantly, although rifampin induces CYP 450 isoenzymes potentially lowering efavirenz exposure.”...

Author Response

Response to Reviewers' Comments:

Reviewer 2 

  1. Comments and Suggestions for Authors

Tuberculosis is one of the leading opportunistic infections and the most important cause of death in patients with human immunodeficiency virus (HIV) infection, despite near-universal access to antiretroviral therapy (ART) and tuberculosis preventive therapy. For patients with active tuberculosis but not yet receiving ART, starting ART after anti-tuberculosis treatment can complicate clinical management due to drug toxicities, drug-drug interactions, and immune reconstitution inflammatory syndrome (IRIS) events. Over the years, there has been significant progress in ART-related research for individuals who are coinfected with HIV and M. tuberculosis. 

The purpose of this review was to help doctors in treating patients who are co-infected with both HIV and tuberculosis. The authors collected and analyzed the most recent research on antiretroviral therapy (ART) for people with HIV-associated tuberculosis. The review is well-written and organized, and the tables are clear and informative, providing valuable insights into the latest treatment guidelines for patients with HIV/TB. The topics cover the major challenges in treating patients with HIV/TB.

Response: We thank the reviewer for the positive feedback and meaningful comments as follows that help us improving the quality of our manuscript.

Comment 1: It is recommended to have a native English speaker review the manuscript to ensure its accuracy and clarity. For instance, on page 8, item 5, line 32, the sentence reads "Importantly, although rifampin induces CYP 450 isoenzymes potentially lowing efavirenz exposure", when it should read "Importantly, although rifampin induces CYP 450 isoenzymes, potentially lowering efavirenz exposure.”

Response: Thank for this valuable and constructive suggestion. We have consulted a native English speaker to optimized our language expression to guarantee the precision and lucidity. The polished modifications have been made in the revised paper.

Comment 2: It is essential that the authors adopt a consistent terminology to describe HIV infection throughout the manuscript. This will prevent any confusion and promote clarity. They should use a single term, such as "people living with HIV/AIDS," and avoid using "HIV-positive people" or "HIV-infected individuals.”

Response: We are very appreciated for this wise advice. We have used the uniform terminology—People living with HIV (PLWH)—to describe HIV infection throughout the revised paper.

Comment 3: I suggest the authors include the following sentence in their writing: "IRIS has been linked to various pathogens and autoimmune diseases, but mycobacterial infections are the most common cause of IRIS." The way it is written, it seems that IRIS is only associated with HIV/TB co-infection. Still, many opportunistic infections, such as fungal, viral, and parasitic infections, can also be linked to IRIS.

Response: We are grateful for your enlightening proposition. The sentence has been added in the revised manuscript.

Comment 4: The authors discuss unmasking TB-IRIS in just four lines (135-138) in their review article. However, I believe that they should elaborate more on this definition of IRIS. For instance, they can include information about the percentage of individuals who develop this type of IRIS and the risk factors associated with the emergence of unmasked IRIS.

Response: Thanks for the constructive suggestion. We have added related contents to further elaborate the definition and clinical character of unmasked IRIS and provided the percentage data of unmasking IRIS reported in two studies (item 6.1). Because some risk factors of unmasked and paradoxical IRIS are overlapped and several factors are difficult to classify as belonging to unmasked IRIS or paradoxical IRIS. In that case, we summarized all potential risk factors comprehensively in item 6.2 (including 6.2.1, 6.2.2 and 6.2.3). We hope the revision will be suitable.

Comment 5: The authors discussed the predictive factors for IRIS in lines 163-166. However, some additional factors, such as the antigenic load, the degree of immunological restoration after cART treatment, and the host's genetic susceptibility, are also important and should be included. It's worth noting that these mechanisms can interact with each other, leading to the development of the syndrome.

Response: We thank the reviewer for this comment. We have added some other factors enclosed with corresponding references—including high viral load, the degree of immunological restoration after cART treatment, Black ethnic origin and elevated extrapulmonary extracellular matrix turnover— in the revised manuscript (item 6.2.1).

  1. Comments on the Quality of English Language

It is recommended to have a native English speaker review the manuscript to ensure its accuracy and clarity. For instance, on page 8, item 5, line 32, the sentence reads "Importantly, although rifampin induces CYP 450 isoenzymes potentially lowing efavirenz exposure", when it should read "Importantly, although rifampin induces CYP 450 isoenzymes potentially lowering efavirenz exposure.”...

Response: Thanks for your good suggestions. We have consulted a native English speaker and made the according modifications in the revised paper.